# UNCERTAINTY-GUIDED LIFELONG LEARNING IN BAYESIAN NETWORKS

## ABSTRACT

Sequentious learning of tasks arriving in a continuous stream is a complex problem and becomes more challenging when the model has a fixed capacity. Lifelong learning aims at learning new tasks without forgetting previously learnt ones as well as freeing up capacity for learning future tasks. We argue that identifying the most influential parameters in a representation learned for one task plays a critical role to decide on *what to remember* for continual learning. Motivated by the statistically-grounded uncertainty defined in Bayesian neural networks, we propose to formulate a Bayesian lifelong learning framework, `BLLL`, that addresses two lifelong learning directions: 1) completely eliminating catastrophic forgetting using weight pruning, where a hard selection mask freezes the most certain parameters (`BLLL-PRN`) and 2) reducing catastrophic forgetting by adaptively regularizing the learning rates using the parameter uncertainty (`BLLL-REG`). While `BLLL-PRN` is by definition a zero-forgetting guaranteed method, `BLLL-REG`, despite exhibiting some small forgetting, is a task-agnostic lifelong learner, which does not require to know when a new task arrives. This feature makes `BLLL-REG` a more convenient candidate for applications such as robotics or on-line learning in which such information is not available. We evaluate our Bayesian learning approaches extensively on diverse object classification datasets in short and long sequences of tasks and perform superior or marginally better than the existing approaches.

## 1 INTRODUCTION

Humans can easily accumulate and maintain the knowledge gained from previously observed tasks and continuously learn to solve new problems or tasks. Inspired by this capability of humans, Lifelong Learning (LLL) intends to learn sequential tasks without forgetting Unlike the powerful capabilities observed in mammalian brains in acquiring and preserving the knowledge, artificial agents still struggle in continual learning.

One of the main challenges for a lifelong learning agent is known as *catastrophic forgetting* (McCloskey & Cohen, 1989; McClelland et al., 1995) which refers to the significant drop in the performance of a learner when switching from a trained task to a new one. This phenomenon occurs because trained parameters on initial task require to change in favor of learning new objectives. This is the reason that naive finetuning intuitively suffers the most from *catastrophic forgetting*. Given the limited capacity of a network, one way to see this problem as to deciding which part of the network is best to keep and which part is less important and can thus be used by future tasks without significantly affecting current task performance. In this work, we propose to turn to Bayesian Networks as introduced by Blundell et al. (2015). These networks represent each parameter with a distribution defined by a mean and variance which more naturally allows determining which parameters are most important for a task. We integrate this idea with two different LLL paradigms: Hard pruning of part of the network and soft regularization based importance weighting of network parameters.

To avoid adding capacity when learning new tasks, one can first prune or compress the network (Aghasi et al., 2017) and then learn new tasks using free parameters as proposed by (Mallya & Lazebnik, 2017; 2018). Here a part of the network is frozen, while the remaining network is free to adapt for future tasks. While this eliminates forgetting over time, it reduces flexibility when learning

large number tasks as the network fills up over time as the percentage which is frozen by previous tasks increases over time.

In non-Bayesian lifelong learning approaches, model parameters tend to overconfidently fit training data and there is no internal definition of weight importance. In contrast, Bayesian models, enhanced by their statistically-grounded theoretical framework, provide clear definitions of uncertainty in model weights and/or predictions. Although exact Bayesian inference remains intractable in neural networks, variational approximations have provided a powerful alternative to this problem. In this work we exploit an efficient back-propagation-compatible Bayesian formulation in neural networks, called Bayes-by-Backpop (BBB) Blundell et al. (2015), to obtain weights' uncertainty as a guidance to perform continual learning. The main intuition behind is that Bayesian neural network provide precise uncertainty measurements in weights as a by-product which can be used as an accurate notion of importance to perform lifelong learning.

## 2 RELATED WORK

Here we list some of the previously introduced approaches taken in lifelong learning that mitigate catastrophic forgetting to different extents. Conceptually, these approaches can be divided into the following categories. We list pros and cons for each and explain why (not) our approach is inspired or build upon them.

**Dynamic architectural methods**: In this setting, the architecture will grow such that it carries over the past knowledge as well as storing new gained knowledge in different forms such as additional layers, nodes, or dual-architecture. In this approach, the objective function remains fixed whereas the model capacity grows often exponentially with the number of tasks. Progressive networks Rusu et al. (2016); Schwarz et al. (2018) was one of the earliest work in this path which was successfully applied on reinforcement learning problems where the base architecture is duplicated and lateral connections are added in response to new tasks. Roy et al. (2018) also uses hierarchical network, where CNN's at multiple levels, could grow in tree-shape manner to accommodate new tasks. Dynamically Expandable Network (DEN) Yoon et al. (2018) also expands the network by selecting the *drifting* units and retraining them separately based on the current tasks in hand. Despite the ability of these methods in preventing forgetting, they have large overheads upon arrival of new tasks. We have completely avoided using architectural growth in our method due to this reason.

**Memory-based methods:** In this regime, previous information is partially stored in different fashions to be used later as a form of *rehearsal* Robins (1995). iCaRL Rebuffi et al. uses this idea to perform class incremental learning through nearest-mean-of-exemplars using current data and stored examples of previously seen classes which was shown to scale up to incrementally learning ImageNet. However, this method can not be used in learning multiple sequential tasks from different datasets. Gradient episodic memory (GEM) (Lopez-Paz et al., 2017) was another work for class incremental learning where data is stored at the end of each episode to be used later to restrict gradient updates from deviating away from their previous values. GEM is not only able to mitigate forgetting but also allows for positive backward knowledge transfer which is referred to improvement on previously learned tasks. Another value of this work was that it was the first method capable of learning with seeing each training example only once. Since storing examples can be costly and sometimes unpractical, a generative model was employed to encode/decode pseudo information Shin et al. (2017); Nguyen et al. (2017). However, in practice, generative models are difficult to train especially in sequential learning scenarios.

**Regularization methods**: These approaches do not require storing data while retaining the whole network. Catastrophic forgetting is alleviated via preventing significant changes to representation learned for previous tasks. This can be performed through regularizing the objective function or directly applied on weight parameters such that important weights to one task are not allowed to change in the future. Typically, this *importance* measure is engineered to represent the importance of each parameter. Inspired by Bayesian learning, elastic weight consolidation (EWC) method (Kirkpatrick et al., 2017) important parameters are those to have the highest in terms if the Fisher information metric. In Synaptic Intelligence (SI) (Zenke et al., 2017) this parameter importance notion is engineered to correlate with the loss function, parameters that contribute more to the loss are more important. Similar to SI (Zenke et al., 2017), Memory-aware Synapses (MAS) Aljundi et al. (2017) proposed an online way of computing importance adaptive to the test set using the change in

the model outputs w.r.t the inputs. PackNet Mallya & Lazebnik (2017) used iterative pruning to fully restrict gradient updates on important weights via saving binary masks. While applying this binary mask at inference results in zero forgetting of the initial performance, it requires to know which task we are performing to use the appropriate mask. PackNet also ranks the weight importance by their magnitude which is not guaranteed to be a proper importance indicative. HAT (Serrà et al., 2018) identifies important neuron using an attention vector to the task embedding to control the gradient flow through neurons.

# 3 BACKGROUND: VARIATIONAL BAYES-BY-BACKPROP

In this section, we review the Bayesian framework we use to be able to learn sequential tasks by remembering task-specific parameters. The Bayesian formulation used here was first introduced by Blundell et al. (2015) that learns the probability distribution over network parameters. In the original paper, it was shown that this novel back-propagation-compatible algorithm acts as a regularizer and yields comparable performance to dropout on MNIST dataset. In the next section (Section 4) we show how to use the derived weight uncertainty to measure the importance in performing the task in hand and use the rest of the network to learn later tasks with.

## 3.1 VARIATIONAL INFERENCE IN NEURAL NETWORKS

In Bayesian models, latent variables are drawn from a prior density $p(\mathbf{w})$ which are related to the observations through the likelihood $p(x|\mathbf{w})$. During the inference, the posterior distribution $p(\mathbf{w}|\mathbf{x})$ is computed conditioned on the given input data. However, in practice, this probability distribution is intractable and is often estimated through approximate inference. Markov chain Monte Carlo (MCMC) sampling (Hastings, 1970) has been widely used and explored for this purpose (see Robert & Casella (2013) for different methods under this category). However, MCMC algorithms, despite providing guarantees in finding asymptotically exact samples from the target distribution, they are not suitable for large datasets and/or large models as they are bounded by speed and scalability issues. Alternatively, Variational inference provides a faster solution to the same problem in which the posterior is approximated using optimization rather than being sampled from a chain Peterson (1987); Hinton & Van Camp (1993); Jaakkola & Jordan (1996; 1997). Similar to any other approximation algorithms, there will be trade-offs between the precision of the results and the efficiency of the optimization. However, variational inference methods always take advantage of fast optimization techniques such as stochastic methods, distributed methods, which allows them to explore data models faster to find the best approximation. See Blei et al. (2017) for a complete review of the theory.

## 3.2 BAYES BY BACKPROP

Let $\mathbf{x} \in \mathbb{R}^n$ be a set of observed variables and $\mathbf{w}$ be a set of latent variables parametrized by $\theta$ sharing a joint density of $p(\mathbf{w}, \mathbf{x})$. Given the observations $\mathbf{x}$ and by using the set of weight parameters $\mathbf{w}$, a neural network, as a probabilistic model, $p(\mathbf{y}|\mathbf{x}, \mathbf{w})$ can output a probability associated with a possible output $\mathbf{y} \in \mathcal{Y}$. Variational inference aims to calculate this conditional probability distribution over the latent variables by finding the closest proxy to the exact posterior through solving an optimization problem.

We first propose a family $\mathcal{D}$ of probability densities over the latent variables ($q(\mathbf{w}|\theta) \in \mathcal{D}$). We then find the closest member of this family to the true conditional probability of interest $p(\mathbf{w}|\mathbf{x})$ by minimizing the Kullback-Leibler (KL) divergence between $q$ and $p$:

$$q^*(\mathbf{w}|\theta) = \underset{q(\mathbf{w}|x)\in\mathcal{D}}{\arg\min} \mathrm{KL}\big(q(\mathbf{w}|\theta)\|p(\mathbf{w}|\mathbf{x})\big) \tag{1}$$

Once solved, $q(\cdot)$ would be the closest approximation to variational posterior parametrized by $\theta^*$ to construct the parameters of the network. We can unpack Eq. 1 and call it a loss function commonly known as variational free energy or expected lower bound :

$$\mathcal{L}_{BBB}(\mathbf{w}, \theta) = \mathrm{KL}\big[q(\mathbf{w}|\theta)\|p(\mathbf{w})\big] - \mathbb{E}_{q(\mathbf{w}|\theta)}\big[\log(p(\mathbf{x}|\mathbf{w})\big] \tag{2}$$

Blundell et al. (2015) showed that Eq. 2 can be approximated using $i$ Monte Carlo samples from the variational posterior:

$$\mathcal{L}_{BBB}(\mathbf{w}, \theta) \approx \sum_{i=1}^{n} \log q(\mathbf{w}^{(\mathbf{i})}|\theta) - \log \mathbf{p}(\mathbf{w}^{(\mathbf{i})}) - \log p(\mathbf{x}|\mathbf{w}^{(i)}) \qquad (3)$$

We assume $q(\mathbf{w}|\theta)$ to have a Gaussian pdf with diagonal covariance. $\theta$ is parametrized point-wise with a mean $\mu$ and standard deviation of $\sigma = \log(1 + \exp(\rho))$. Variational posterior parameters will have a form of $\mu + \log(1 + \exp(\rho)) \circ \epsilon$ where $\epsilon$ is a sample drawn from posited $q$. For the prior, as it was suggested in Blundell et al. (2015), a scale mixture of two Gaussian pdfs are chosen which are zero-centered while having difference variances of $\sigma_1^2$ and $\sigma_2^2$. The uncertainty obtained for every weight parameter, as opposed to hidden units Rezende et al. (2014); Kingma & Welling (2013) has been successfully used in model compression Blundell et al. (2015), uncertainty-based exploration in reinforcement learning application Blundell et al. (2015). In this work we want to use this framework to learn sequential tasks without forgetting using per-weight derived uncertainties.

## 4 LIFELONG LEARNING USING BAYES BY BACKPROP (BLLL)

In our method called, Bayesian Lifelong Learning (BLLL), we have considered two common scenarios of lifelong learning: 1) sequential learning of multiple datasets; 2) class-incrementally learning of a single or two alternating datasets.

Having uncertainty measured precisely for mean and variances of our posterior distributions, one can choose two paths to perform continual learning; 1) catastrophic forgetting elimination and 2) catastrophic forgetting mitigations where in the former, called BLLL-PRN, important weights are kept frozen using a binary pruning mask saved per task whereas, in the latter called BLLL-REG, important parameters are regularized using an adaptive learning rate during gradient update in which the a learning rate of a parameter scales with its uncertainty. We will explain BLLL-PRN and BLLL-REG in details in section 4.1 and 4.2.

### 4.1 BLLL WITHOUT FORGETTING USING WEIGHT PRUNING (BLLL-PRN)

Unlike regular deep neural networks, in BBB model, weight parameters are represented by probability distributions parametrized by their mean and standard deviation. In order to take into account both mean and variance, we use the quantity of signal-to-noise ratio (SNR) for each parameter defined as

$$\text{SNR} = \frac{|\mu|}{\sigma} \qquad (4)$$

SNR is a commonly used measure in signal processing to distinguish between "useful" information from unwanted noise contained in a signal. In the context of neural models, SNR can be thought as an indicative of parameter importance; the higher the SNR, the more effective or important the parameter is to the model predictions for a given task.

Weight pruning is a long standing solution to reduce inference computation or network compression (Liu et al., 2017; Molchanov et al., 2016). Moreover, it has been recently used to perform lifelong learning (Mallya & Lazebnik, 2017), where the goal is to sequentially learning multiple tasks using a single network's capacity. In (Mallya & Lazebnik, 2017) this has been accomplished by freeing up parameters deemed to be unimportant to the current task according to their magnitude. Forgetting is prevented in pruning by saving a task-specific binary mask of important vs. unimportant parameters. Our work is significantly different from (Mallya & Lazebnik, 2017) in the different criterion we propose to use for importance measurement; the statistically-grounded uncertainty defined in Bayesian neural networks.

BLLL-PRN is performed as follows: for every layer, convolutional or fully-connected, the parameters are ordered by their SNR value and those with the lowest importance are pruned (set to zero). The pruned parameters are "saved" using a binary mask so that they can be used later in learning new tasks whereas the important parameters remain fixed throughout training on future tasks. Once a task is learned, an associated binary mask is saved which will be used at inference to recover key parameters to the desired task.

---

**Algorithm 1** Bayesian Lifelong Learning with Zero Forgetting (`BLLL-PRN`) or Reduced Forgetting (`BLLL-REG`)

---

**Require:** hyper parameters for BBB-only: $\sigma, \sigma_1, \sigma_2, \rho, \mu, \pi, n$
**Require:** hyper parameters for training: learning rate $\alpha$

1: $\epsilon \sim \mathcal{N}(0, I)$
2: $\sigma = \log(1 + \exp(\rho))$        $\triangleright$ Ensures $\sigma$ is always positive
3: $\mathbf{w} = t(\theta, \epsilon) = \mu + \sigma \circ \epsilon$       $\triangleright$ $\mathbf{w} :=$ a posterior sample of weights
4: **while** not done **do**
5:   $l_1 \leftarrow \sum_{i=1}^{n} \log \mathcal{N}(\mathbf{w}_i | \mu, \sigma^2)$       $\triangleright$ log-posterior
6:   $l_2 \leftarrow \sum_{i=1}^{n} \log\left(\pi \mathcal{N}(\mathbf{w}_i \mid 0, \sigma_1^2) + (1 - \pi)\mathcal{N}(\mathbf{w}_i \mid 0, \sigma_2^2)\right)$    $\triangleright$ log prior
7:   $l_3 \leftarrow \sum_{i=1}^{n} \log(p(x|\mathbf{w}))$       $\triangleright$ log-likelihood of the data
8:   $\mathcal{L}_{BBB} \leftarrow \frac{1}{M}(l_1 - l_2) - l_3$      $\triangleright$ $M :=$ number of minibatches
9:   **if** BLLL-PRN **then**
10:    $\text{SNR}_i \leftarrow \frac{|\mu_i|}{\sigma_i}$       $\triangleright$ signal-to-noise ratio for every weight
11:    **for all** layers **do**
12:     **for all** parameters in layer **do**
13:      **if** $\text{SNR}_i \in$ Top $p$ portion of SNR values in layer **then**
14:       $\alpha = 0$
15:    $\mu' \leftarrow \mu - \alpha \nabla \mathcal{L}_{BBB_\mu}$      $\triangleright$ Update parameter using gradient descent
16:    $\rho' \leftarrow \rho - \alpha \nabla \mathcal{L}_{BBB_\rho}$      $\triangleright$ Update parameter using gradient descent
17:   **if** BLLL-REG **then**
18:    $\beta \leftarrow \sigma \times \alpha$
19:    $\mu' \leftarrow \mu - \beta \nabla \mathcal{L}_{BBB_\mu}$
20:    $\rho' \leftarrow \rho - \alpha \nabla \mathcal{L}_{BBB_\rho}$    $\triangleright$ Update the remaining uniformly using gradient descent

---

Similar to (Mallya & Lazebnik, 2017) the overhead memory per parameter in encoding the mask as well as saving it on the disk is as follows. Assuming we have $n$ tasks to learn using a single network, the total number of required bits to encode an accumulated mask for a parameter is at max $\log_2 n$ bits assuming a parameter deemed to be important from task 1 and kept being encoded in the mask. Saving the binary mask for our a typical model with $n$ tasks results in a mask size of $1/n^2$ with respect to the initial size.

## 4.2 BLLL WITH REDUCED FORGETTING (`BLLL-REG`)

Another strategy to perform continual learning is to reduce forgetting by controlling further changes in a representation learned for a task. This can be enforced in a form of regularization in the objective function or through a varying learning rate for weight parameters.

In order to prevent drastic changes to the predicted posterior for a learned task, we propose to condition the changes in the mean of each parameter's distribution on its uncertainty i.e., the more certain the mean of a distribution is, the less it should be allowed to be updated to learn future concepts with. We impose this through scaling the learning rate for $\mu_i$ with its uncertainty represented by $\sigma_i$. The intuition behind this is to develop a gradient regularizer that adjust the gradient updates using the existing uncertainty in the means. In other words, we wish to prevent any further changes on the certain mean by lowering their learning rate at each gradient update while its variance is allowed to change. This results in allowing the model learning more concepts while preserving the critical information obtained in the past.

The key benefit of `BLLL-Reg` variant using learning rate as the regularizer is that it neither requires additional memory, as opposed to pruning technique nor tracking the change in parameters with respect to the previous learned task, as needed in common weight regularization methods. More importantly, this method does not need to be aware of task switching as it only needs to adjust the learning rates of the means in the posterior distribution based on their current standard deviations. Complete algorithm for `BLLL` with pruning (`BLLL-PRN`) and adaptive learning rate (`BLLL-REG`). is shown in Algorithm 1.

## 5 EXPERIMENTAL SETUP

**Datasets:** Toy benchmarks such as MNIST(LeCun et al., 1998) and CIFAR10/100 (Krizhevsky & Hinton, 2009) have been widely used to measure forgetting and accuracy of various lifelong learning approaches. As a result, we include them as part of our evaluation. We use MNIST split and permuted MNIST (Srivastava et al., 2013) for class incremental learning with similar experimental settings used in Serrà et al. (2018); Lopez-Paz et al. (2017) in terms of data splits. Furthermore, to have a better understanding of our method, we also evaluate our approach on sequentially learning of various datasets belonging to different distributions similar to the sequence of 8 tasks used in (Serrà et al., 2018) including FaceScrub (Ng & Winkler, 2014), MNIST, CIFAR100, NotMNIST (not), SVHN (Netzer et al., 2011), CIFAR10, TrafficSigns (Stallkamp et al., 2011), and FashionMNIST (Xiao et al., 2017). Details of each are summarized in Table 1. In all the experiments we resized images to $32 \times 32 \times 3$ if necessary. For datasets with monochromatic images, we replicate the image across all RGB channels. No data augmentation of any kind has been used in our analysis.

Table 1: Utilized datasets summary

| Names | #Classes | Train | Test |
|---|---|---|---|
| FaceScrub (Ng & Winkler, 2014) | 100 | 20,600 | 2,289 |
| MNIST (LeCun et al., 1998) | 10 | 60,000 | 10,000 |
| CIFAR100 (Krizhevsky & Hinton, 2009) | 100 | 50,000 | 10,000 |
| NotMNIST (not) | 10 | 16,853 | 1,873 |
| SVHN (Netzer et al., 2011) | 100 | 73,257 | 26,032 |
| CIFAR10 (Krizhevsky & Hinton, 2009) | 10 | 39,209 | 12,630 |
| TrafficSigns (Stallkamp et al., 2011) | 43 | 39,209 | 12,630 |
| FashionMNIST (Xiao et al., 2017) | 10 | 60,000 | 10,000 |

**Baselines:** Within Bayesian framework, we have three reference baselines of fine-tuning, feature extraction, and joint training. In fine-tuning (`BLLL-FT`), training with regular SGD optimization continues upon arrival of new tasks without any forgetting avoidance strategy. Feature extraction, denoted as (`BLLL-FE`) in our experiments, refers to freezing all layers in the network except for the last layer when a new task arrives. In joint training (`BLLL-JT`) we learn all the tasks in a multi-task learning fashion which serves as the upper bound for average accuracy on all tasks. We have compared these three Bayesian references with their counterparts using an ordinary (non-Bayesian) network referred to as `ORD-FT`, `ORD-FE`, and `ORD-JT`. We have compared them on Split MNIST class incremental learning experiments. Along with the BLLL variants of reference baselines, we compare with the state-of-the-art approaches including EWC Kirkpatrick et al. (2017), IMM (Lee et al., 2017), learning without forgetting (LWF) (Li & Hoiem, 2017), less-forgetting learning (LFL) (Jung et al., 2016), PathNet (Fernando et al., 2017), PNNs (Rusu et al., 2016), and Hard attenstion mask (HAT) (Serrà et al., 2018) using implementations provided in (Serrà et al., 2018). On Permuted MNIST dataset average accuracies are directly reported from GEM (Lopez-Paz et al., 2017), SI (Zenke et al., 2017), and VCL (Nguyen et al., 2017) without re-implementation.

**Network architecture:** For Split MNIST and Permuted MNIST experiments, we have used a multilayer perceptron with three hidden layers with 660 units whereas other baselines use a two-layer MLP with 800 units in each. Because there exists double number of parameters in our bayesian network compared to its equivalent regular neural net, we ensured fair comparison by matching the total number of parameters between the two to be 1.9M unless otherwise is stated. For multiple datasets learning scenario as well as alternating incremental CIFAR10/100 datasets we have used a ResNet18 Bayesian network with 7.1-11.3M parameters depending on the experiment. Note that the majority of the baselines provided in this work were originally developed using some variants of AlexNet structure. We tried change that to ResNet type of network, however, it resulted in degrade in their reported and experimented performance. On the other hand, best results for our Bayesian network were achieved using ResNet18. Therefore, we kept the architectures as is and only matched their number of parameters to ensure having equal capacity across different approaches.

**Hyperparameter tuning:** We have used the approach introduced in (Anonymous, 2019) specific to continual learning setting for hyperparameter tunning. Unlike commonly used tuning techniques which use a validation set composed of all the classes in the dataset, this method realistically assumes to have access to a few classes only on which we can tune the hyperparameters on a held-out validation set. In all our experiments we consider 0.15 split for validation set on the first two tasks in all the learning sequences.

**Training details:** It is important to note that in all our experiments, no pre-trained model is used. We used stochastic gradient descent with learning rate of $0.01$ decaying by a factor of $0.3$ upon being plateau for $5$ consecutive epochs. Batch size of $64$ was consistently used throughout all experiments. Dataset splits and batch shuffle has been performed identically across BLLL experiments and all the baselines.

**Performance measurement:** Let $n$ be the total number of tasks. Once all are learned, we evaluate our model on all $n$ tasks. ACC is the average test classification accuracy across all tasks. To measure forgetting we report backward transfer, BWT, which indicates how much learning new tasks has influenced the performance on previous tasks. While $\text{BWT} < 0$ directly reports *catastrophic forgetting*, $\text{BWT} > 0$ indicates that learning new tasks has helped with the preceding tasks. In mathematical notation BWT and ACC are as follows:

$$\text{BWT} = \frac{1}{n-1} \sum_{i=1}^{n-1} R_{i,n} - R_{i,i}, \quad \text{ACC} = \frac{1}{n} \sum_{i=1}^{n} R_{i,n} \tag{5}$$

where $R_{i,n}$ is the test classification accuracy on task $i$ after sequentially finishing learning the $n^{\text{th}}$ task.

## 6 RESULTS & DISCUSSION

We present our results for class incremental learning of single datasets (Split MNIST and Permuted MNIST) as well as sequential learning of $8$ distinct tasks using variant of BLLL approach compared against various recent lifelong learning approaches.

### 6.1 SPLIT MNIST

We first present our results for class incremental learning of MNIST (Split MNIST) in which we learn $0 - 9$ digits in two tasks with $5$ randomly shuffled classes at a time. All presented methods in this setup have $1.9M$ parameters. Figure 2a shows the results for reference baselines in Bayesian and non-Bayesian networks including fine-tuning (BLLL-FT,ORD-FT), feature extraction (BLLL-FE,ORD-FE) and, joint training (BLLL-JT,ORD-JT) averaged over $5$ runs. Although MNIST dataset is too easy to draw broad conclusions on it, we observe throughout all experiments that Bayesian fine-tuning and joint training, perform significantly better than their counterparts ORD-FT and ORD-JT that use a regular network where forgetting in BLLL-FT is comparable and even better than some of the baselines presented in literature. We also compare against HAT (Serrà et al., 2018) and LWF (Li & Hoiem, 2017) where HAT has similar performance to our BLLL-PRN with zero forgetting while BLLL-REG is able to increase the ACC by $2\%$ at the cost of $0.1\%$ drop in forgetting.

Figure 3a shows how the accuracy carries from the first half of the dataset to the second half. Ordinary fine-tuning is the only reference method that experience a performance boos upon change in the task whereas both BLLL-FE and ORD-FE degrade after new task arrives which is fully expected. All other baselines, as well as our BLLL invariants, steadily perform above $\approx 99\%$ on both tasks.

**Pruning procedure:** Here we explain how we choose the pruning portion in general with an example using Split MNIST. In continual learning we cannot assume we know how many more tasks we yet have to learn, therefore, we can only decide based on the current performance on validation set. By obtaining the drop in validation accuracy as a function of pruning $\%$, as shown in Fig. 1, we will be able to choose the pruning $\%$ value depending on the difficulty of the dataset by setting a threshold

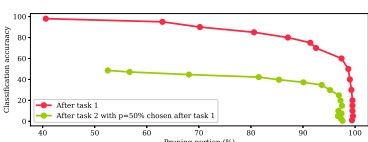

Figure 1: Validation accuracy as a function of pruning percentage for Split MNIST

beyond which we do not wish to prune because of the significant degrade in the performance. In this example we chose to prune by $50\%$ after learning task 1 which clearly results in limiting our capacity if we were to learn more than two tasks with this model.

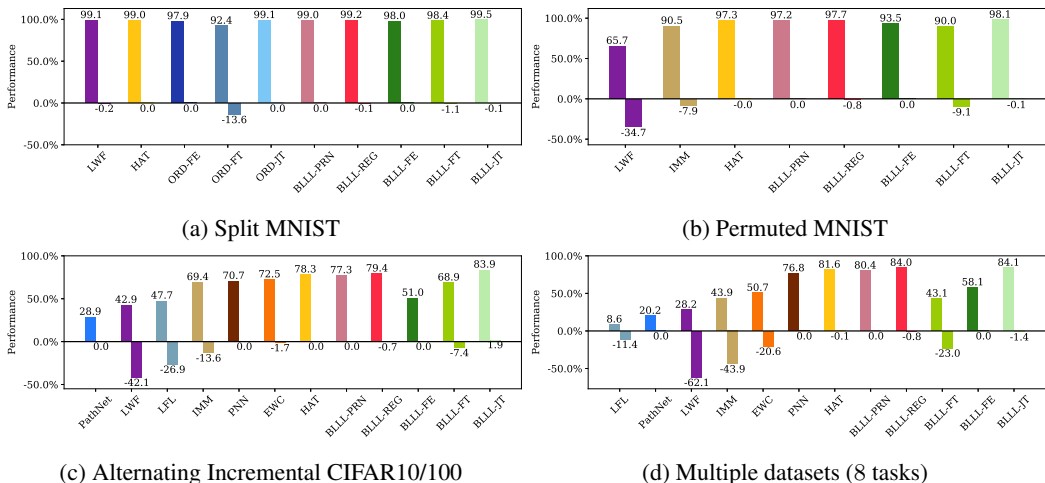

(a) Split MNIST

(b) Permuted MNIST

(c) Alternating Incremental CIFAR10/100

(d) Multiple datasets (8 tasks)

Figure 2: ACC and BWT on (a) Split MNIST (2 tasks), (b) Permuted MNIST (10 tasks), (c) Alternating incremental CIFAR10/100 (10 tasks), (d) 8 consecutive tasks shown in Table 1

## 6.2 PERMUTED MNIST

Permuted MNIST is a popular variant of MNIST dataset to evaluate continual learning approaches in which each task is considered as a random permutation of the original MNIST pixels. Following the literature, we learn a sequence of 10 random permutations and report average accuracy at the end. Figure 2b shows ACC and BWT performance of `BLLL-REG` and `BLLL-PRN` against that of the state of the art models. We can again observe that `BLLL-FT`, despite having no intention in preventing forgetting, exhibits reasonable negative BWT values which makes it a successor over IMM (Lee et al., 2017) and LWF (Li & Hoiem, 2017) baselines. While HAT (Serrà et al., 2018) and `BLLL-PRN` have a similar ACC performance with no forgetting, `BLLL-REG` shows a slight increase in the tendency to forget ($-0.8\%$) in exchange for a negligible boost in ACC ($0.4\%$). Accuracy evolution over 10 tasks is shown in 3b.

Results in Fig. 2b are computed on a 1.9M network. However ACC for this experiment is also reported in HAT (Serrà et al., 2018), VCL (Nguyen et al., 2017), EWC (Kirkpatrick et al., 2017), SI (Zenke et al., 2017), and GEM(Lopez-Paz et al., 2017) using a smaller MLP with 0.1M parameters. ACC for these approaches are given as $91.0\%$, $90\%$, $88.2\%$, $86.0\%$, $82.6\%$, respectively. However, `BLLL-REG` achieves $92.2\%$ with BWT of $-0.4\%$ under the same conditions.

## 6.3 ALTERNATING CIFAR10 AND CIFAR100

In this experiment, we alternate between class incrementally learning of CIFAR10 and CIFAR100. Both datasets are divided into 5 tasks each with 2 and 20 classes per task, respectively. Figure 2c presents ACC and BWT obtained with `BLLL-PRN`, `BLLL-REG`, and three `BLLL` reference methods compared against various continual learning baselines. It is interesting to note that in this setup, some baselines (LWF and PathNet) do not perform better than the naive accuracy achieved by feature extraction. LFL is able to marginally outperform feature extraction in ACC at the cost of twice of the forgetting we can have in fine-tuning while IMM performs only as well as fine-tuning in both ACC and BWT. PNN, EWC, and HAT are the only baselines that perform between the minimum expected ACC (`BLLL-FE`) and maximum acceptable negative BWT (`BLLL-FT`) bounded by the high ACC and near zero forgetting achieved in joint training (`BLLL-JT`) for this setup.

Among the three acceptable baselines, PNN is the only zero-forgetting-guaranteed approach and hence can be properly compared with our `BLLL-PRN` which also never forgets by definition. Note that PathNet is also a zero-forgetting approach which suffers from bad pre-assignment of network's capacity per task which causes poor performance on the beginning task from which it never recovers. `BLLL-PRN` consistently outperforms PNN by overall $10\%$ better ACC where the gap between them is more pronounced on CIFAR100 tasks which are more difficult compared to binary classification

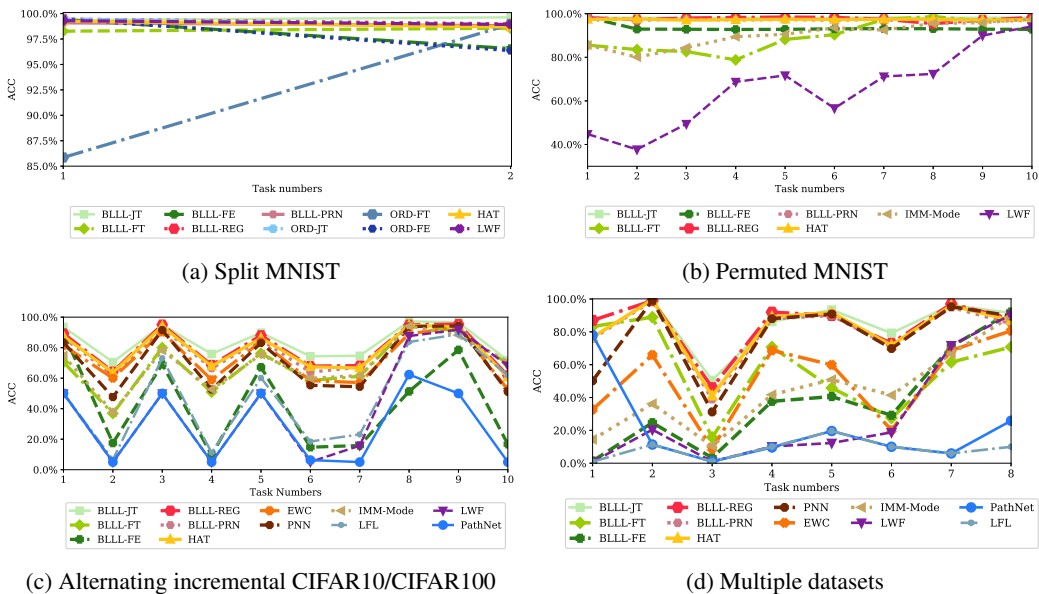

(a) Split MNIST

(b) Permuted MNIST

(c) Alternating incremental CIFAR10/CIFAR100

(d) Multiple datasets

Figure 3: Evolution of the test classification accuracy obtained across tasks when all tasks are learning

tasks of CIFAR10. EWC and HAT are both allowed to forget by construction, however, HAT shows zero forgetting behavior. While EWC is outperformed by both of our BLLL variants, HAT exhibits $1\%$ better ACC over `BLLL-PRN`. `BLLL-REG` is able to compensate for that at a cost of $0.7\%$ drop in forgetting. A trade off between ACC and BWT exists when choosing between `BLLL-PRN` and `BLLL-REG` depending on what is needed in the problem.

## 6.4 MULTIPLE DATASETS LEARNING

Finally, we present our results for sequentially learning of 8 tasks using `BLLL-PRN` and `BLLL-REG`. Similar to the previous experiments we look at both ACC and BWT obtained for `BLLL-PRN`, `BLLL-REG`, BLLL references (`FT,FE,JT`) as well as various baselines. Considering the ACC achieved by `BLLL-FE` or `BLLL-FT` ($58.1\%$) as a lower bound we observe again that some baselines are not able to do better than `BLLL-FT` including LFL, PathNet, LWF, IMM, and EWC. PNN and HAT remain the only considerable baselines for our `BLLL-PRN` and `BLLL-REG` approaches. `BLLL-PRN` continues to outperform PNN by $5\%$ in ACC while the two ensure zero forgetting. HAT exhibits $-0.1\%$ BWT while being overcome by `BLLL-REG` in ACC by $3\%$.

We believe the main insight of our Bayesian approach is that instead of computing extra measurements of importance, which are often task, input or output dependent, we are able to use the predicted weights uncertainty as a bi-product of the framework to deem the most contributing parameters in the overall posterior distribution and choose to freeze them using a binary mask, as in `BLLL-PRN`, or to enforce minimum changes in them conditioned on their current uncertainty, as in `BLLL-REG`, allowing for more capacity to learn with and most importantly developing a task-agnostic model that does not require to be warned of task arrival. This makes our approach an excellent candidate for applications in which such information is inevitably unavailable.

## 7 CONCLUSION

In this work we propose a lifelong learning formulation for Bayesian networks, called `BLLL`, that uses the Bayesian-embedded uncertainty predictions as a guidance to perform lifelong learning; i.e. to identify important parameters that can be either fully preserved through a saved binary mask (`BLLL-PRN`) or allowed to change conditioned on their uncertainty for learning new tasks (`BLLL-REG`). We demonstrated how the probabilistic uncertainty distributions per weights are helpful in sequentially learning short and long sequences of benchmark datasets compared against many other baselines. We showed that both `BLLL-PRN` and `BLLL-REG` perform superior or marginally

better than the state-of-the-art models such as HAT (Serrà et al., 2018) across all the experiments. In order to choose between the two `BLLL` variant the followings should be considered: `BLLL-PRN` results in zero forgetting while requiring a small binary mask to be saved per task whereas `BLLL-REG` is memory-free and allows for more capacity in the network by allowing small forgetting to occur. The key benefit from `BLLL-PRN` is that the model no longer need to know when tasks are being switched leading to a task-agnostic model that can be deployed where it is not possible to distinguish tasks in a continuous stream of the data.

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
