# OpenReview forum: "Uncertainty-guided Lifelong Learning in Bayesian Networks"
_ICLR.cc/2019/Conference_

### Official Review · AnonReviewer3 · 2018-11-02
**Nice combination of ideas, but requires more development.**

**Rating:** 4
**Confidence:** 4

**Review:**

In this paper, a framework for lifelong learning based on Bayesian neural network is proposed. The key idea is to combine iterative pruning for multi-task learning along with the weight regularization. The idea of iterative pruning was first considered by Mallya et al., 2018 and weight regularization was considered for Bayesian neural network by Nguyen et al., 2018.

Pros:
- Combination of two idea seems novel. I like the idea of considering the weight parameter as the "global" random variables and the mask parameters as the task-specific random variables.

Cons:
- In general, there is lack of explanation/justification on the combination of two ideas. Especially, there is lack of explanation on how to apply the whole algorithm (e.g., text states that complete algorithm is in Algorithm 3., but there is no Algorithm 3. in the paper).

- I do not understand how equation (6) is developed, and why hyper-parameters are need for "regularization of weights", comparing with the Variational Continual Learning (VCL, Nguyen et al., 2018). More explanation seems necessary for justification of the algorithm.

- More stronger baselines need to be considered for the experiments. Why is there no comparison with the existing continual learning algorithms? At the very least, comparison with the VCL or Elastic Weight Consolidation (EWC, Kirkpatrick et al., 2017) seems necessary since one of the key idea is about regularization for weights.


In general, I think it is a nice idea to combine two existing approaches. However, the algorithm lacks justification in general and experimental results are not very persuasive.

---

> ### Author Response · Authors · 2018-11-27
> **Responses to individual comments from R#3**
>
> We would like to thank you again for your time and your feedback. Please see first our meta response to major comments shared between reviewers. Here we refer to separate comments/questions we received from you.
>
> We also found it behaving inconsistent through the new experimental setting we adapted per reviewers' request. Therefore, as an alternative, we used a simple regularization trick on the learning rates instead of directly minimizing the changes on network parameters. Please see our updated draft regarding the altered regularization method.
>
> We have tried to address this feedback provided by all reviewers by replicating the experimental setting used in the literature to be able to make fair comparisons. Specifically asked by R#1, we have a full comparison with 7 other baselines and 8 datasets. Please see the updated version of the paper regarding this matter.

---

### Official Review · AnonReviewer2 · 2018-11-02

**Rating:** 4
**Confidence:** 4

**Review:**

The paper addresses the problem of lifelong learning of neural networks - a setting where learning is performed on a continuously arriving new tasks without having access to previously encountered data.
Authors propose a method that prevents catastrophic forgetting typical for naive application of stochastic gradient descent by preventing supposedly important weights to change (in either soft of hard manner), where the weight importance is assessed by its signal to noise ratio estimated from the corresponding (approximate) posterior distribution.
Authors evaluate their approach on a set of image classification datasets and find it superior to the PackNet baseline as well as few simpler ones.

The idea of using uncertainty estimates obtained from Bayesian training to adjust weight updates is natural and potentially very promising.
However, to me this paper does not seem to investigate the idea sufficiently deep.

The weight pruning or hard masking variant of the method depends on a very important hyperparameter p (size of the mask) which is unclear how to set beforehand.

I also struggle with understanding the weight regularisation or soft masking variant.
Authors seem to get their inspiration in the idea of assumed density filtering, where the posterior for 1:T-1 is approximated and used a prior for task T (last sentence on page 5).
At the same time, in Algorithm 2, line 6 the prior is defined as the standard BBB mixture prior and not the approximate posterior from the previous task.
Quite oddly, parameters of the _approximate posterior_ are being quadratically regularalized to not deviate from parameters of the _approximate posterior_ from the previous task.
This deviates from the original idea and requires additional justification.
Besides that, I find the way this regularisation is applied potentially problematic for the variance parameter (last term in eq. 6).
Here authors apply the regularisation to the parameter of the softplus transformation they use, but scale it with the inverse std deviation which is the “classical” parametrisation. The choice of parametrisation was not discussed, however, clearly different parametrizations may lead to very different results.

On the experimental side, I have two major issues:
1. The datasets considered are very small, authors could consider using ImageNet, especially given that they already work with 224x224 images.
2. The only prior work used as a baseline is PackNet, while there is no reason why other established methods such as EWC are not applicable.

Minor comments:
The middle expression in eq. 5 seems to miss the -log p(D_T | D_{1:T-1}) term which does not change the latter expression (since it does not depend on parameters theta).
Page 3: “citestochastic methods”, a citation seems to be missing.

---

> ### Author Response · Authors · 2018-11-27
> **Responses to individual comments from R#2**
>
> We would like to thank you again for your time and your feedback. Please see first our meta response to major comments shared between reviewers. Here we refer to separate comments/questions we received from you.
>
> We acknowledge all your concerns regarding our proposed regularization technique. We also found it behaving inconsistent through the new experimental setting we adapted per reviewers' request. Therefore, as an alternative, we used a simple regularization trick on the learning rates instead of directly minimizing the changes on network parameters. Please see our updated draft regarding the altered regularization method.
>
> Pruning percentage: this is a valid concern which we have now addressed in Figure 1 and a subsection under section 6.1. We briefly explain here that in the continual learning regime we do not know how many tasks we yet have to learn. Therefore, we can only decide based on the current performance of our model on a held-out validation set. By computing validation accuracy as a function of pruning percentage we can set a threshold beyond which, we do not wish to downgrade in our performance. Figure 1 shows such a plot for MNIST Split dataset when incrementally learned in two tasks.
>
> Your concern regarding the datasets size and number of baselines are now addressed in the updated draft with having 7 baselines with 8 short and long sequences of tasks
>
> The listed typos and minor issues are now fixed.

---

### Official Review · AnonReviewer1 · 2018-11-03
**Good motivation, minor contributions in term of algorithms**

**Rating:** 4
**Confidence:** 4

**Review:**

Motivated from leveraging the uncertainty information in Bayesian learning, the authors propose two algorithms to prevent forgetting: Pruning and Regularization. Experiments on several sequential learning tasks show the improved performance.

Quality:  The description on the related work is comprehensive. The proposed algorithms seem easy to follow.

Clarity: Low

The contributions in terms of algorithms are clearly presented. However, the writing can be largely improved.

(1) Some claims are improper:  I don't think it's accurate to say that most of lifelong learning is non-Bayesian (In introduction), and EWC is derived from a Bayesian perspective, and Variational Conditional Learning is a very Bayesian approach.

(2) Please proofread the submission:
Typos: e.g.,  "Beysian", "citestochastic methods";
Style: x is not bold occasionally, but has the meaning given the context.

Originality: It seems to be the first work that leverages the variance in Bayesian Neural Nets (BNN) to prevent forgetting. My understanding that EWC also consider the variance, but in a different way.

Significance:
It is good to consider variance/uncertainty for lifelong learning, and should be encouraged.
However, the comparison to the representative algorithms or state-of-the-art is missing in this submission. For example, EWC/IS, or method in [*].  Is it possible to run the experiments on more standard datasets, such as [*].

[*] Overcoming Catastrophic Forgetting with Hard Attention to the Task, ICML 2018


Questions:
1. In (6), there are three terms on the right side, it seems the 2nd term include the 3rd term, why do we need to add the 3rd term again?

2. "Once a task is learned, an associated binary mask is saved which will be used at inference to recover key parameters to the desired task. The overhead memory caused by saving the binary mask (less than 20MB for ResNet18), is negligible given the fact it completely eliminates the forgetting"

To me, saving a binary mask means saving "partial" model. First, this is additional parameter saving. Second, in the inference stage, one can recover the corresponding best model using the mask, how close is it to cheating? (Perhaps I am not an expert in lifelong learning).
Can you put the model size of ResNet18, so that the readers can understand 20MB is small/negligible compared to the full model.

---

> ### Author Response · Authors · 2018-11-27
> **Responses to individual comments from R#1**
>
> Please see the addressed comments shared between reviewers first. Here we refer to separate comments/questions we received from you:
>
> (1) We have corrected this in the manuscript (page 2 paragraph 2)
>
> (2) Fixed.
>
> (3)  We would like to thank you again for suggesting the mentioned paper by Serrà, Joan, et al. (2018). Per your request, we have changed our experimental setting in accordance to it and included full comparison with the datasets provided in [*]. Paper is fully updated with the obtained results.
>
>
> Questions you raised:
>
> (1) We have modified our regularization approach explained in our shared meta response. Instead of minimizing the changes between current parameters and updated values, we scale up or down their learning rate conditioned on how important they are, i.e. how big their STD is. Hence equation 6 no longer exists in the paper.
>
> (2) This is a valid point and we agree that the memory size overhead was not clearly explained so let us clarify this with explaining how much encoding a mask and writing it to memory will cost us. The overhead memory per parameter in encoding the mask as well as saving it on the disk is as follows. Assuming we have $n$ tasks to learn using a single network, the total number of required bits to encode an accumulated mask for a parameter is at max $\log_2{n}$ bits assuming a parameter was found to be important from task $1$ and kept being encoded in the mask. Saving the binary mask for a typical model with $n$ tasks results in a mask size of $1/n^2$ with respect to the initial model size.
>
>
> *** Per your request on improving the text we have re-written large parts of the text.

---

### Author Response · Authors · 2018-11-27
**Addressing comments shared between reviewers**

We thank all the reviewers for their constructive feedback and time. We would like to address some common concerns across all the reviewers first before going to individual responses.

1- All reviewers had fairly asked for more experiments and baselines, usage of larger datasets and deeper analysis.

We believe this was a valid point which we have tied to address as much as we can. Upon Reviewer #1’s request we have used similar experimental setting introduced in (Serrà, Joan, et al. 2018) and compared against 7 baselines including HAT, EWC, PathNet, PNN, LWF, LFL, IMM on short and long sequences of 8 datasets in total including Split MNIST, Permuted MNIST, Alternating incremental CIFAR10/100  FaceScrub, Not NotMNIST, SVHN, TrafficSigns, and FashionMNIST.
We have also included reference baselines such as fine-tuning and feature extraction, as well as joint training using both Bayesian and non-Bayesian networks.

Upon Reviewer#3’s request  we have also compared against VCL, as well as GEM, and IS on Permuted MNIST. Due to the extensive evaluations, we fully switched to the tasks provided in Serrà, Joan, et al. 2018 and abandoned fine grained classification datasets we had in the initial version.

2. All reviewers had comments and questions regarding the regularization method. While experimenting with the new datasets with our Bayesian approach, we came to realize that the regularization method which was initially introduced in our paper exhibits inconsistent behavior to overcome forgetting on different datasets, leading us to believe it is not a promising approach. Instead, we were able to find an alternative simpler regularization technique that is also easier to comprehend and empirically performs better.

 The change in the regularization method is as follows: instead of minimizing the change in both mean and variance of the parameters distributions, we now control the gradient updates for mean of the distributions based on the predicted uncertainty we have for them. This mean that we begin with a usual constant learning rate for all parameters, and as we train for more epochs, we compute sigma (standard deviation) of the means. We simply used the STD as an indicative of their uncertainty. The more uncertain (higher STD) a parameter is computed to have, the more it should be allowed to be updated in future epochs. Therefore, we use uncertainty (STD) as a scalar to scale up or down the learning rate of all Mu parameters, The intuition behind this is that we wish to minimize any further changes on the means by simply imposing a small learning rate to them while allowing the variances to change. This results in allowing the model to learn more concepts while preserving the critical information obtained in the past. We used this intuitive regularization trick throughout the paper when we were not using pruning.
The key benefit from using such an approach is that we do not need to wait for a task to finish to find  the most important parameters. We do not even need to know when tasks switching occurs. By simply modifying our optimizer to adjust the learning rate based on the computed uncertainty, we regularize at every epoch, resulting in a model that is less prone to forget.

Paper has been updated with all these changes and added experiments.

---

### Meta-Review · Area_Chair1 · 2018-12-15

**Confidence:** 2
**Recommendation:** Reject

**Metareview:**

Reviewers are in a consensus and recommended to reject. However, the reviewers did not engage at all with the authors, and did not acknowledge whether their concerns have been answered. I therefore lean to reject, and would recommend the authors to resubmit. Please take reviewers' comments into consideration to improve your submission should you decide to resubmit.